# Diet as a Source of Acrolein: Molecular Basis of Aldehyde Biological Activity in Diabetes and Digestive System Diseases

**DOI:** 10.3390/ijms24076579

**Published:** 2023-03-31

**Authors:** Pawel Hikisz, Damian Jacenik

**Affiliations:** 1Department of Oncobiology and Epigenetics, Faculty of Biology and Environmental Protection, University of Lodz, ul. Pomorska 141/143, 90-236 Lodz, Poland; 2Department of Cytobiochemistry, Faculty of Biology and Environmental Protection, University of Lodz, ul. Pomorska 141/143, 90-236 Lodz, Poland

**Keywords:** acrolein, diabetes, diet, diabetic retinopathy, diabetic nephropathy, alcoholic liver disease, colorectal cancer, oxidative stress, DNA adducts, α,β-unsaturated aldehydes

## Abstract

Acrolein, a highly reactive α,β-unsaturated aldehyde, is a compound involved in the pathogenesis of many diseases, including neurodegenerative diseases, cardiovascular and respiratory diseases, diabetes mellitus, and the development of cancers of various origins. In addition to environmental pollution (e.g., from car exhaust fumes) and tobacco smoke, a serious source of acrolein is our daily diet and improper thermal processing of animal and vegetable fats, carbohydrates, and amino acids. Dietary intake is one of the main routes of human exposure to acrolein, which is a major public health concern. This review focuses on the molecular mechanisms of acrolein activity in the context of its involvement in the pathogenesis of diseases related to the digestive system, including diabetes, alcoholic liver disease, and intestinal cancer.

## 1. Introduction

Acrolein (prop-2-enal) was first identified in 1839 by the Swedish chemist J.J. Berzelius, who studied the chemical breakdown of glycerol. Berzelius, a pioneer of biological chemistry, named this substance for its acrid, suffocating odor and physicochemical properties resembling oil (oleum). Interestingly, acrolein’s combination of reactivity, volatility, and toxicity, especially towards respiratory cells, led to the testing and use of acrolein as a chemical warfare agent during World War I [1].

As all such aldehydes, this organic chemical compound from the group of aldehydes (the simplest possible unsaturated aldehyde) is relatively unstable. The structural characteristics of this molecule indicate that the proximity of the electron-withdrawing carbonyl group to the unsaturated bond gives acrolein a strong electrophilicity and, thus, an extremely high reactivity with nucleophilic molecules. This is undoubtedly of significant toxicological importance for acrolein due to its readiness to react with tissue macromolecules [1,2]. Of course, such properties of acrolein have serious consequences for health and the pathogenesis of many diseases in which this aldehyde is involved. Acrolein is responsible for the induction and development of many diseases of the nervous system and cardiovascular system, as well as cancer. The participation of acrolein in the pathogenesis of these diseases is related to the high reactivity of the aldehyde and the ability to interact directly with DNA and numerous proteins involved in redox balance, signal transmission, or the regulation of gene expression [2,3]. The documented mechanisms of acrolein’s molecular activity and cytotoxicity are very different, and the harmful effects of the aldehyde are realized at different levels of cell activity and homeostasis [2,4]. An analysis of the scientific evidence allows us to conclude that the key to the acrolein-dependent pathogenesis of many diseases is the extremely high ability of the aldehyde to generate reactive oxygen species (ROS) and, consequently, to induce oxidative stress [5,6]. In addition, numerous studies indicate that acrolein, by affecting the expression and activity of pro-inflammatory factors, causes a strong immune response, which often leads to chronic inflammation and cell death [7,8]. The effect of acrolein on the key signaling pathways for cell proliferation/death is also significant for the pathogenesis of many diseases [9,10]. Many studies indicate that the mutagenic properties of acrolein result from its significant effectiveness in its direct interaction with DNA and the formation of adducts, primarily with deoxyguanosine [11,12]. In addition, this aldehyde, while interacting with DNA, simultaneously weakens or completely disables repair mechanisms [13].

Importantly, the sources of human exposure to acrolein can be very diverse. The first basic source of it may be endogenous production by enzyme-mediated oxidation of polyamines or threonine, as well as oxidation of unsaturated fatty acids with reactive oxygen species [4]. The main endogenous sources of acrolein are the myeloperoxidase-mediated degradation of threonine and the spermine oxidase-mediated degradation of spermine and spermidine [3,14]. They can be an important source of acrolein in situations of oxidative stress and inflammation. As emphasized by Kashiwagi and Igarashi [14] in their latest work, acrolein formed with the participation of spermine oxidase as a metabolite of spermine biochemical transformations is the main source of tissue damage in the elderly [14]. Acrolein can also be found in air pollution as a product formed by burning gasoline and diesel fuels, wood, and plastics [4,15]. As recent studies have indicated, cigarette smoke is rich in acrolein, which is a serious problem for modern public health, especially with statistical data on the number of smokers and tobacco users. Notably, recent data have demonstrated that the problem of acrolein generation also applies to e-cigarettes [16,17,18,19,20].

A large amount of research is increasingly focusing on our daily diet as one of the main sources of human exposure to acrolein. It turns out that not only what we eat in our daily diet is important, but also how we prepare our meals. Excessive thermal processing of carbohydrates and fatty acids, as well as consumption of excessive amounts of alcoholic beverages, is a dangerous source of acrolein [15,21]. Undoubtedly, an important issue is searching for natural acrolein scavengers, which are components of our food. Strategies to inhibit the harmful effects of acrolein-dependent oxidative stress and counteract the toxicity of dietary acrolein are challenging to implement [15]. Food ingredients with nucleophilic properties are particularly effective at neutralizing acrolein. The recent research provides valuable data by Uemura et al. [22], who showed that 2-furanmethanethiol, cysteine methyl and ethyl esters, alliin, lysine, and taurine decreased acrolein toxicity. In addition to their direct interaction with acrolein and its inactivation, these compounds indirectly inhibited lipid peroxidation and acrolein-dependent cellular damage [22].

This review paper discusses the molecular mechanisms of acrolein activity in diseases directly related to an incorrect diet from different perspectives. Based on a review of in vitro and in vivo studies, we analyze the role of acrolein in the pathogenesis of diabetes (including retino-neuro- and diabetic nephropathy), alcoholic liver disease, and intestinal cancer.

## 2. Methodology

For this review, the relevant literature was searched using three electronic databases: Google Scholar, PubMed, and Scopus. Each experimental paper or review article cited here was published before 12 March 2023 and written in English. The literature search strategy included the following terms and logical hyphens: “acrolein cytotoxicity,” “acrolein intestinal cancer”, “acrolein in the diet”, “acrolein diabetes”, “acrolein diabetic retinopathy”, “acrolein diabetic nephropathy”, “acrolein diabetic neuropathy”, and “acrolein alcoholic liver damage”. These terms were searched for in the abstract, title, or keywords. The relevance of the articles was assessed based on the analysis of their titles and abstracts. Articles meeting all search criteria were evaluated based on their full text and included in this review.

## 3. Watch What You Eat—Daily Diet as a Source of Acrolein

The most significant sources of acrolein for human health are diet, environmental sources, and endogenous reactions. In addition to tobacco smoke and cigarette smoking, both traditional tobacco products and e-cigarettes, diet, and the consumption of certain foods are the primary routes of high acrolein concentrations in humans living in the U.S. The Environmental Protection Agency has acrolein on its list of high-priority toxic chemicals [15]. The International Program on Chemical Safety of the World Health Organization (WHO) has set a daily intake tolerance of acrolein of 7.5 µg/kg bw/day [23]. The content of acrolein in food products varies greatly, and its final concentration is influenced by the preparation method and the extent of thermal processing of food. Its high concentration and ubiquity in food are associated primarily with baking, frying, and fermentation with the participation of bacteria. Interestingly, acrolein is present in many foods such as fruits, vegetables, dairy products, baked and fried foods, salted pork, fish, and alcoholic beverages, including beer, whiskey, brandy, and wine. It is worth noting that a reliable estimation of acrolein exposure via food or water is practically impossible because acrolein concentrations in food change irregularly during food preparation and consumption, and are dependent on eating habits [24,25,26]. In addition, acrolein present in food during gastronomic processing can be transformed by chemical reactions into other toxic substances. One example of such a reaction is the formation of highly toxic and carcinogenic acrylamide in foods, whose precursor is acrolein [27]. Current data on acrolein content in individual food products are very limited and do not allow for a reliable assessment of exposure to aldehyde. Measurements of acrolein concentration should be carried out, especially in processed food in a ready-to-eat state [23].

An additional risk for people working in gastronomy is the inhalation of acrolein in the air of poorly ventilated kitchen rooms, especially where large amounts of food are fried. As indicated by Ewert et al. [26], apart from glycerin, which is dehydrated as a component of edible oils during thermal processing, releasing acrolein, the main source of aldehyde is also the unsaturated backbone of fatty acids [26]. According to research, the concentration of acrolein in the air of commercial and private kitchens is between 26 and 65 µg per cubic meter [28,29,30]. Although these values are much lower than the accepted standards for exposure to acrolein, it is reasonable to draw attention to kitchen exposure to acrolein regarding the possible impact on human health. Hecht et al. [30] found elevated levels of acrolein-derived metabolites in the urine of non-smoking Asian women who regularly prepare wok-fried foods, indicating lung uptake of acrolein and chronic exposure to this harmful aldehyde [30].

The presence of acrolein in the air in kitchens generated when lipids and carbohydrates are heated at high temperatures is a serious threat to human health. Studies conducted in Hong Kong based on samples taken in 15 different commercial restaurants indicate significant concentrations of acrolein in the air. The annual emissions of acrolein from commercial kitchens alone in Hong Kong are estimated to be around 7.7 tons, while the annual emissions of acrolein from car exhaust are only 1.8 tons. These data highlight the scale of the public health problem in the context of a highly urbanized city and acrolein emissions dependent on broadly understood culinary tourism [28]. The inhalation of acrolein and other often harmful kitchen fumes generated during the cooking and thermal processing of food has health consequences [31]. Subsequent studies indicate a close correlation between the exposure to acrolein, which is genotoxic and mutagenic, and an increased risk of lung cancer. More than 30 years ago, Gao et al. [32] outlined the problem of Chinese female cooks exposed to acrolein from wok cooking. The authors point out that acrolein emissions from cooking, not smoking, are associated with a high incidence of lung cancer in Chinese women [32]. These reports are confirmed by the recent studies by Hecht et al. [30,33]. The authors demonstrated that in women who often cook with a wok and, at the same time, do not smoke and do not drink alcoholic beverages, significantly elevated levels of the acrolein metabolite 3-hydroxypropyl mercapturic acid (3-HPMA) were observed in the urine. These findings and epidemiological studies indicate a strong need to pay more attention to the problem of acrolein exposure in commercial kitchens and the urgent need for effective preventive measures [30,33].

### 3.1. Carbohydrates and Acrolein

Carbohydrates are organic compounds that perform many important functions in the human body. Their task is primarily to provide cells with energy by burning glucose. Many of the carbonyl compounds are formed during the thermal processing of food, and their presence often shapes the original smell or taste of food. Carbohydrates, often one of the main nutrients in many food products, are one of the main sources of acrolein in the diet after thermal processing [34]. The first information on the production of acrolein from carbohydrates dates back to the 1960s, when Byrne et al. [35] suggested that acrolein may be produced in the pyrolysis reaction of hexoses [35]. These reports were confirmed in later years in studies using isotopic labeling, which made it possible to precisely trace the process of glucose pyrolysis and, ultimately, the production of acrolein [36]. The key to aldehyde formation is a fragment of the glucose carbon chain from C4 to C6. The entire formation process is a two-stage process. In the first stage, dehydration occurs between C5 and C3, which results in the cleavage of C3 and C4. In a further step, propene-1,3-diol is formed and, after further dehydration, produces acrolein. However, as the authors indicate, dehydration between C1 and C2 and the retro-Diels–Alder reaction on the same diol also leads to the formation of acrolein [36].

Under the influence of thermal processes and long storage, a series of successive reactions occur in food that reduce sugars and amino acids, peptides, or proteins containing a free amino group, which lead to the formation of a large group of new chemical compounds, including acrolein. The Schiff base formed from sugars and amino acids rearranges into Amadori products, which, as intermediates, participate in the production of hydroxyacetone through the dehydration and cleavage of the 3,4-retro aldol Amadori product. Hydroxyacetone can eventually generate acrolein via 2,3-enolization [37,38,39].

When discussing carbohydrates as a source of acrolein in the diet, cigarettes should be mentioned as one of the greatest factors of acrolein exposure. The concentration of carbohydrates added to cigarettes directly affects the amount of acrolein in cigarette smoke to which smokers are exposed [40]. Recent reports by Pennings et al. [41] showed a strong correlation between the concentration of harmful aldehydes (including acrolein) in cigarette smoke and the concentration of sugars added to cigarettes [41]. As with the case of classic cigarettes, the situation is developing in the recently more and more popular e-cigarettes, whose users are also exposed to high acrolein formed during the combustion of liquid e-cigarette components. As highlighted by several recent studies, the degree of generation of harmful aldehydes (including acrolein) depends on many factors, with the key ones being the presence of propylene glycol (PG) and vegetable glycerin (VG) [42,43,44]. As Vreeke et al. [45] and Samuborowa et al. [46] pointed out, the thermal degradation of VG and PG during the smoking of e-cigarettes is crucial for producing acrolein. These reports are confirmed by the recent, highly valuable work of Fagan et al. [42], who analyzed the composition of 66 different brands of e-cigarettes. One of the conclusions of the obtained results was the correlation between the amount of acrolein generated during e-liquid combustion and the content of sugars [42].

### 3.2. Alcohol and Acrolein

Interestingly, the source of acrolein in the diet can also be drinks, including both non-alcoholic beverages, such as tea and coffee, and those containing alcohol—especially wine and spirits [23]. The consumption of large amounts of alcoholic beverages is a significant source of exposure to acrolein, particularly in the gastrointestinal tract [15,24]. Among other things, acrolein and phenolic compounds shape the bitterness of wine alcohols, which is correlated with alcoholic fermentation and the decomposition of glycerol by bacteria [21,47,48]. As indicated by Alvarenga et al. [21], the presence of acrolein and other compounds such as butan-2-ol, propan-1-ol and volatile acids in sugar cane spirit is associated with the activity of *Lactobacillus* bacteria and reactions involving glycerol [21]. Glycerol can be dehydrated to 3-hydroxy propionaldehyde (3-HPA) by the B12-dependent coenzyme glycerol/diol dehydratase enzyme system and converted to acrolein by another dehydration reaction [49,50]. It is worth noting that 3-HPA is not the end product of alcoholic fermentation and can be further reduced to 1,3-propanediol by an oxidoreductase. Moreover, 3-HPA can exist in both dimeric and hydrated forms. Depending on the pH of the environment, a different amount of 3-HPA oligomers is formed, mainly due to acetalization or the aldol reaction. This mechanism is crucial for the concentration of acrolein in wine products because, depending on the pH or temperature of the environment, 3-HPA and its derivatives, in various forms, are converted into acrolein. The HPA system regulates the final concentration of acrolein. High temperatures and an acidic pH have favored the conversion of 3-HPA to acrolein [48]. As emphasized by Jiang et al. [15], although the presence of acrolein in alcoholic products is often associated primarily with giving them a bitter taste, it is necessary to pay attention to the harmful effects of acrolein in these products for the whole organism. Chronic exposure to acrolein ingested with alcohol can be a serious problem, especially in terms of cytotoxicity to the cells of the mouth and digestive system [15].

### 3.3. Lipids and Acrolein

Undoubtedly, one of the main sources of acrolein in the daily diet is edible oils and fatty acids, which are broken down to aldehyde under high temperatures accompanying the preparation of meals. The fate of the formed acrolein can be very different; some remains in a volatile form and is released into the air, whereas some of the acrolein remains in the heated oil or goes directly into the meal and, during digestion, is absorbed by the cells of the digestive system, posing a threat to humans [3]. The obvious source of acrolein is glycerol, derived from the thermal decomposition of di- and triglycerides. Breaking the ester bond between the OH groups of glycerol and the COOH of fatty acids provides free molecules of fatty acids and glycerol, a precursor of acrolein, in further chemical transformations [51,52]. For years, scientists have wondered whether acrolein can be produced from fats, e.g., as a result of lipid peroxidation with the participation of free radicals. Some authors questioned the possibility of producing acrolein from polyunsaturated fatty acids (PUFAs) in free-radical lipid oxidation [53,54]. The first information on the pathways of acrolein formation from triglycerides in cooked oils and the participation of free-radical lipid peroxidation reactions comes from the late 1990s and the work of Umano et al. [55]. As the authors suggested, acrolein may be formed by successive homolytic fissions of ester linkages [55]. A breakthrough in understanding the source of acrolein from fatty acids and free-radical reactions was achieved with the work by Esterbauer et al. in 1991 [56]. The authors described the formation of acrolein from arachidonic acid. However, as they emphasized, acrolein is not formed either from the terminus of the alkyl fatty acid or from the carboxy terminus (reduction of the R-COOH carboxyl group to the R-CHO carbonyl group does not take place in an oxidative environment) [56].

Acrolein, on the other hand, is formed from the middle part of the aliphatic fatty acid chain (carbon chain). Two reactions are responsible for generating acrolein in the oxidative environment, breaking lipid hydroperoxides’ carbon–carbon bonds: β-cleavage of the corresponding alkoxy radical and Hock fragmentation [56]. The exact chemical mechanism of the reaction is presented in the work of Yin and Porter [57]. The formation of acrolein from the lipid hydroperoxide precursor includes at least one β-cleavage fragment, including an alkyl terminus. In addition, the aldehyde moiety can be formed either by β-cleavage of the intermediate alkoxy radical or by Hock hydroperoxide cleavage [57]. In recent years, subsequent studies have confirmed the assumptions of the work of Esterbauer et al. [56] and the mechanism of generating acrolein from fatty acids. Ewer et al. [26] described the formation of acrolein during the heat processing of oils (linoleic acid) to be similar to that proposed by Esterbauer [56], while Endo et al. [58] indicated that linolenic acid may be one of the main sources of acrolein in heated vegetable oils. Acrolein is formed as a result of hydroperoxide and epidioxide appearing during the thermal processing of vegetable oils. Additionally, the latest research by Kato et al. [59] using linoleic acid and a-linolenic acid emphasizes the key role of reactive oxygen species in forming acrolein from lipid peroxidation products [59].

When discussing fats as a source of acrolein in food, it is reasonable to distinguish between studies conducted with oils (di- or triglycerides) and studies conducted with fatty acids or their methyl esters. Comparative studies conducted by Pederson et al. [60] on the heat-induced release of acrolein (and other volatile substances) from rapeseed oil, rapeseed fatty acid methyl esters, and petroleum diesel revealed differences in the amount of acrolein released. Interestingly, rapeseed oil, after heating, “produced” at least 100 times more acrolein than a mixture of fatty acid methyl esters, which clearly shows how important glycerol is in the formation of acrolein in foods during thermal processing [60]. Several papers have indicated that high levels of acrolein in the air persist in commercial restaurant kitchens [28,61,62]. Umano and Shibamoto [55] evaluated the effect of the temperature and time of heating edible oils and beef fat on the degree of acrolein release. As expected, an increase in acrolein formation was observed in direct proportion to time and temperature. Interestingly, the authors indicated a negative correlation between acrolein production and a measure of the degree of unsaturation of fatty acid groups. The availability of water molecules in the reaction medium limits the production of acrolein from triglycerides. According to the authors’ hypothesis, fatty acid residues with a higher degree of unsaturation are more amenable to adding water, thus using water that would otherwise be available for ester hydrolysis and the release of free glycerol. Notably, significantly greater acrolein formation was observed with oxygen than with nitrogen, indicating that free-radical mechanisms and reactive oxygen species play a significant role in the generation of acrolein [55]. Of course, the source of the generation of acrolein is not only saturated fatty acids but also unsaturated omega acids from fish, which are generally considered healthy and desirable in a balanced daily diet. Recent studies have shown that acrolein is one of the main volatile compounds released during the thermal processing of fish oils, and oxidation plays an important role here. The ω-3 fatty acids eicosatetraenoic acid (EPA) and docosahexaenoic acid (DHA) present in the fish oil are oxidized into hydroperoxide and rapidly converted to an alkoxyl radical and a hydroxyl radical. Further reactions lead to the formation of propanal and an alkyl radical. Acrolein is formed as a result of the subsequent reaction of this alkyl radical with hydroxyl radicals [63,64].

## 4. Acrolein in the Pathogenesis of Diabetes

Diabetes mellitus is currently a global public health challenge as it is the most common endocrine disorder. The progressive increase in morbidity had caused modern researchers to increasingly point to its epidemic character, and the World Health Organization (WHO) considers diabetes to be one of the most serious threats of the 21st century [65,66]. Diabetes is defined as a disease characterized by elevated blood glucose levels. Its causes are complex, and symptoms may appear suddenly or only after several years. Diabetes belongs to a group of metabolic diseases characterized by hyperglycemia resulting from defects in insulin secretion and/or action. Chronic hyperglycemia in diabetes is associated with damage and functional disorders as well as the failure of various organs, such as the eyes, kidneys, urinary system, heart, and blood vessels [67]. Diabetes patients are particularly at risk of complications resulting from sugar metabolism disorders. The most dangerous include diabetic retinopathy (DR) and diabetic nephropathy (DN). These diseases, as one of the late complications of diabetes, are associated with structural and functional changes in the kidneys and retina of the eye, caused directly by hyperglycemia. They damage the organs mentioned above and ultimately can result in their complete dysfunction, which threatens the health and life of patients [68,69]. Numerous studies indicate a large share of acrolein in the pathogenesis of many diseases related to cardiovascular diseases [70], neurodegenerative diseases [71], and chronic obstructive pulmonary disease (COPD) [5], as well as in the carcinogenesis of various types of cancer [9,72]. Accumulated evidence also clearly indicates the participation of acrolein in the development of diabetic complications, such as DR, DN, and neuronal dysfunction in patients with sugar metabolism disorders [4,73,74].

Although the exact mechanism responsible for the participation of acrolein in the pathogenesis of diabetes is not fully understood, more and more evidence indicates its significant role in this complex metabolic disease. The molecular basis of acrolein’s involvement in the pathogenesis of diabetes is shown in Figure 1. Studies have shown a significant correlation between the concentrations of acrolein and its intermediate metabolites in both type 1 and type 2 diabetes [75,76,77,78]. A recent study by Feroe et al. [73] on diabetic samples from 2027 adults who participated in the 2005–2006 National Health and Nutrition Examination Survey (NHANES) showed elevated levels of the acrolein metabolite 3-HPMA in the blood of patients is correlated with the stage of diabetes, glycemic status, and insulin resistance [73].

The molecular mechanisms of acrolein activity in the pathogenesis of diabetes and related diseases are implemented in many biological pathways through various events. Analyzing the available literature and a number of in vitro and in vivo studies conducted on both mammalian cell models and rodent models, three basic aspects of acrolein activity seem to be of key importance [4,74,79]. Acrolein, as a highly reactive molecule, interacts with numerous proteins, leading to their carbonylation and damage, a significant reduction in the level of antioxidants, including that of L-gamma-Glutamyl-L-cysteinyl-glycine (glutathione; GSH), and to damage to the mitochondria and oxygen metabolism. The result of these actions of acrolein is the disruption of redox homeostasis in cells and the generation of strong oxidative stress, which is crucial for many diseases, including diabetes [79]. Of course, acrolein-dependent oxidative stress has further molecular targets. Other factors that are not without significance in diabetes and the accompanying DN, DR, and neurodegeneration include acrolein-dependent chronic inflammation, damage to blood vessels, the dysfunction of Müller cells, and the effect on cell division and proliferation [80,81,82,83].

### 4.1. Acrolein-Dependent Oxidative Stress Generation: Damage to Proteins and Mitochondria

The main mechanism that causes acrolein to be so dangerous to human health is its unusually high reactivity and ability to interact with a number of proteins. Protein–acrolein adducts can cause significant structural and, thus, functional modifications of proteins. One of the most common protein–acrolein adducts is the interaction between aldehyde and lysine to form Nε-(3-formyl-3,4-dehydropiperidino) lysine (FDP-Lys). FDP-Lys is used in research as a biomarker of acrolein and to assess the body’s exposure to acrolein [84,85,86]. Numerous in vitro and in vivo studies in diabetic models in patients and rodents have indicated an elevated level of FDP-Lys of various proteins (both serum and hemoglobin) and a strong correlation between their concentration and the severity and further progression of diabetic effects such as retinopathies, neuropathies, or nephropathy [74,77,82,87,88]. Acrolein-dependent protein carbonylation is considered one of the main ways to generate oxidative stress in cells [89]. As a result of protein carbonylation with the participation of acrolein, it is possible to produce carbonyl species and increase oxidative damage. Evidence suggests that such events are characteristic of diabetes, particularly in diabetic retinopathy or nephropathy [74,90,91]. Of course, the main target of the attack for acrolein remains reactive oxygen species (ROS)-scavenging proteins, which are directly involved in maintaining the redox balance in our cells [2,3,92,93]. The etiology of diabetes and its accompanying DR and DN is multifactorial, and an important element is oxidative stress. The role of ROS in the pathogenesis of diabetes has been extensively investigated in numerous studies [94,95].

Since thiol compounds are particularly attractive targets for attack by acrolein, GSH is one of the most significant proteins to be inactivated by acrolein [1,2]. The mechanism of acrolein-dependent GSH depletion is the basic factor leading to the generation of oxidative stress in diabetic patients. As the research emphasizes, acrolein caused a decrease in GSH activity in retinal pigment epithelium (RPE) cells [91,96], retinal microvascular endothelial cells, and glial cells [88,92], leading to the consequent increase in oxidative stress and cell death. Moreover, acrolein caused a significant decrease in the expression of the glutamate–cysteine ligase (GCL) enzyme, which is involved in the control of GSH production [91]. Of course, GSH is not the only target of attack for acrolein in terms of its involvement in generating oxidative stress in diabetes. As evidenced by research conducted on RPE cells, the decrease in the total antioxidant capacity was correlated with the acrolein-dependent inactivation of superoxide dismutase (SOD), intracellular glutathione peroxidase, or glutathione S-transferase (GST) [91,96]. In addition, as research by Murata et al. [88] and Wu et al. [97] indicates, an important role in generating oxidative stress in diabetes, including both limiting the antioxidant defense system and increasing ROS generation, is played by spermine oxidation with the participation of vascular adhesion protein-1 (VAP-1) and spermine oxidase (SMOX). Acrolein, formed in the process of spermine oxidation, participates through the continuous generation of FDP-Lys adducts and the generation of ROS in the pathogenesis of DR during diabetes [88,97].

The inevitable consequence of the protein carbonylation, the oxidative stress generated by acrolein, and the production of aldehyde-dependent free radicals is the mitochondrial damage characteristic of the pathogenesis of diabetes [98,99]. Studies on RPE cells proved that acrolein caused mitochondrial dysfunction by changing the potential of the mitochondrial membrane and increasing intracellular Ca^2+^ concentration. Moreover, the exposure of cells to acrolein led to disturbances in the energy management of organelles, manifested by a significant reduction in the activity of mitochondrial complexes (I, II, and V) and ATP depletion [91,96,100,101].

The high affinity between acrolein and various proteins is likely also responsible for aldehyde’s ability to modulate the expression of glucose transporter type 4 (GLUT4) [102]. The impairment of glucose transport in skeletal muscle is one of the main factors responsible for reducing glucose uptake in type 2 diabetic patients [103]. The exposure of mice to acrolein significantly increased fasting blood glucose levels and impaired glucose tolerance, which was caused by decreased GLUT4 expression in rodent muscles. The molecular basis of these events was the acrolein-dependent phosphorylation suppression of key signaling proteins for GLUT4 expression and carbohydrate metabolism, including insulin receptor (IR) substrate 1 (IRS1), protein kinase B (Akt), mammalian target of rapamycin kinase (mTOR), ribosomal protein S6 kinase beta-1 (p70S6K), and glycogen synthase kinase 3 α/β (GSK3 α/β) [102].

### 4.2. Acrolein Cytotoxicity: Neurodegeneration, Vascular Damage and Nephrotoxicity

The biological consequence of the strong interaction of acrolein with proteins and their carbonylation is often cell death by apoptosis due to the impairment of its key life functions [74,81,83]. During the development of diabetes, acrolein-dependent nephrotoxicity, neurodegeneration, damage to epithelial and vascular cells, and accompanying disorders of blood vessel integrity are particularly important. A growing body of evidence indicates that both neurons, nephrons, and microvascular endothelial cells are particularly exposed to the cytotoxic effects of acrolein. Moreover, the death of these cells underlies the pathogenesis of diabetes and is involved in many accompanying diseases [74,77,82,91,101,104].

Elevated levels of FDP-Lys and immunomodulatory properties of acrolein often accompany its cytotoxicity. Liu et al. [104] reported elevated levels of FDP-Lys in the ganglion cell layer of the retina in a diabetic animal model. The level of adducts was correlated with, e.g., a reduction in retinal function [104]. Similar biological effects of the formation of acrolein–protein conjugates have been observed in in vivo and in vitro studies in kidney cells [74]. Acrolein–protein adducts were negatively correlated with kidney function. An increase in the activity of caspase-9 and -3, the degradation of poly(ADP-ribose) polymerase-1 (PARP-1), and the activation of apoptosis were observed [74].

As described by Dong et al. [82], FDP-Lys conjugates can accumulate in endothelial cells and pericytes, and these events are characteristic of the course of DR. As the authors emphasized, the resulting adducts contribute to disturbances in the integrity of blood vessels and significant changes in the viability of retinal microvascular endothelial cells (HRMECs) [82]. These data may be supplemented by the work of Murata et al. [88], who indicated that acrolein-dependent GSH depletion in the rat retinal capillary endothelial cell line (TR-iBRB2) and the induction of oxidative stress are other important factors leading to cell death, and thus, impaired the blood–retina barrier integrity during diabetes [88].

Ferroptosis is a relatively new type of regulated iron-dependent cell death associated with free-radical lipid peroxidation and the apoptosis-independent caspase cascade [105]. Cell death by ferroptosis occurs through the inhibition of glutathione peroxidase 4 (GPX-4) due to increased iron levels in the cell. This enzyme is responsible for removing free radicals; however, its activity significantly decreases during ferroptosis. As a result of too much ROS of fatty acids, membrane damage occurs, leading to cell death. Ferroptosis has been found in Parkinson’s, Alzheimer’s, and neurodegenerative diseases [106,107]. The latest research by Qi et al. [108] provides extremely interesting information, as they used the zebrafish research model to assess the neurodegenerative properties of acrolein and its impact on the pathogenesis of diabetic peripheral neuropathy. Exposure to acrolein resulted in impaired motor neuron development. Moreover, acrolein treatment induced the ferroptosis of nerve cells. However, as the authors emphasized, the precise mechanism of acrolein-dependent induction of this type of cell death is not fully understood, and the potential pathways and molecular targets involved in ferroptosis and peripheral neurogenesis need further study [108].

### 4.3. Immunomodulatory Properties of Acrolein and Diabetes

As the research emphasizes, acrolein, a component of cigarette smoke, has a permanent immunomodulatory effect [7,8,109,110]. Reports of acrolein-dependent changes in inflammatory signaling and the expression of genes involved in the immune response are mixed. Burcham et al. [1] emphasized that the factors influencing the immunogenic properties of acrolein are its dose, the degree of exposure of the host cell, and its type. Immunomodulatory properties of acrolein are correlated with the aldehyde-dependent control of the expression of pro-inflammatory factors and regulators of gene expression, especially interleukins, including IL-8 and nuclear factor κB (NF-κB), and tumor necrosis factor α (TNF-α) [7,8,109,110].

Indeed, the role of acrolein in modulating the immune response in the pathogenesis of diabetes and its complications is not as well studied as in the cases of cigarettes and COPD. However, several pieces of evidence provide clear evidence of acrolein involvement in immunomodulatory disorders and in diabetes mellitus, especially in relation to DN and DR [74,77,81,83]. As pointed out by McDowell et al. [77], rats exposed to acrolein showed an increase in Müller cell gliosis in conjunction with the upregulation of the receptor for advanced glycation end products (RAGE) and calcium-binding protein B (S100B). These events resulted in the increased secretion of, among others, pro-inflammatory mediators IL-1β and intercellular adhesion molecule 1 (ICAM-1) [77]. As emphasized by Murata et al. [92], who showed an increase in the expression of the pro-inflammatory chemokine C-X-C motif chemokine ligand 1 (CXCL1) in their studies using rat retinal Müller cell line TR-MUL5, the oxidative stress induced by acrolein may play a key role in shaping inflammation in diabetes [92]. Clear evidence of the role of acrolein in DR inflammation is provided by Grigsby et al. [83], who showed an increase in the activation of transforming growth factor beta-1 and overproduction of vascular endothelial growth factor (VEGF) in ARPE-19 retinal cells, and consequently, cell viability reduction [83]. These reports are consistent with in vivo studies in rats [81], in which the upregulation of inflammatory cytokine expression of IL-1β and TNF-α was observed after exposure to acrolein. An acrolein-dependent increase in pro-inflammatory factors was associated with diabetes-induced neuropathic pain (DNP), which, as the authors emphasized, indicates that acrolein might be involved in the development of neuroinflammation and behavioral consequences of DNP [81]. Very valuable information is also provided by recent in vitro and in vivo studies by Tong et al. [74], who emphasized the main role of acrolein as the culprit in the pathogenesis of DN. Acrolein, in both animal models and two human kidney cell lines (HK2 and HEK293), activated the RAS protein and downstream MAPK pathways, increasing inflammatory cytokines (IL-6, IL-1β, IL-18, and TNF-α) and leading ultimately to severe damage to kidney cells and their apoptosis [74].

### 4.4. Acrolein-Dependent Activation of Müller Glial Cells

Numerous studies among diabetic patients have emphasized the role of glial cells in the pathophysiology of the early stages of diabetic retinopathy. The functioning of the retina is based on the interaction between glial cells (Müller cells and astrocytes) and blood vessels and neurons in the retina. Changes in the metabolism of glial cells disturb retinal homeostasis, and the resulting neuronal damage has a negative impact on small blood vessels [111,112]. It has been proven that the activation of glial cells is one of the initial events occurring in the early stages of DR [113,114]. Hyperglycemia increases hypoxia-inducible factor-1 (HIF-1) and insulin-like growth factor-1 (IGF-1) in both the serum and the vitreous, leading to hypoxia, inflammation, and the consequent activation of glial cells. Activated glial cells secreting pro-inflammatory cytokines create a chronic inflammatory environment closely related to retinal fibrosis. Additionally, phenotypically altered, activated glial cells upregulate VEGF and fibroblast growth factor, thereby inducing retinal fibrosis and pathological neovascularization during DR [114,115,116].

Recent data from in vitro and in vivo studies using diabetic research models indicate that acrolein is crucial in triggering retinal glial cell activation in diabetic eyes. The key to the acrolein-dependent activation of Müller cells seems to be the accumulation of acrolein adducts with FDP-Lys proteins in them and the accompanying oxidative stress and modulation of the expression of specific genes [77,87,92]. Acrolein in the immortalized rat retinal Müller cell line TR-MUL5 caused significant induction of the inflammatory C-X-C motif chemokine ligand 1 (CXCL1) protein, which is a member of a family of chemokines involved in promoting neutrophil migration by binding to the CXCR2 receptor [92]. Importantly, as indicated by previous studies, the concentration of CXCL1 was significantly elevated in the vitreous fluids of patients with DR [117]. Additionally, as indicated by Yong et al. [87], acrolein–protein adducts resulted in the Müller glia of diabetic rats upregulating VEGF transcription. Overexpression of this factor is associated with angiogenesis, the regulation of vascular permeability, and neovascularization during DR pathogenesis [87].

Valuable information on acrolein-dependent glial cell migration and the pathological hallmark of diabetic retinopathy is provided by the latest research from the research team of Fukutsu et al. [80,118]. In their research on rat retinal Müller glial cell line TR-MUL5, they focused on the key retinal glial cell migration and activation of rho-associated coiled-coil-containing protein kinases (ROCKs) [80,118]. Activation of the ROCK-1 signaling pathway induces, among others, focal vascular constrictions, endoluminal blebbing, and subsequent retinal hypoxia [119]. Exposure of TR-MUL5 cells resulted in GSH depletion and the induction of oxidative stress. As emphasized in these studies, a significant increase in the expression of the ROCK1 isoform and activation of the kinase following acrolein stimulation was observed. The consequence of increased ROCK1 activity was that acrolein dependence enhances phosphorylation of myosin phosphatase target subunit 1 (MYPT1) and myosin light-chain 2 (MLC2) through the ROCK1 cascade induced by oxidative stress [80,118].

## 5. The Role of Acrolein in the Pathogenesis of Alcoholic Liver Disease

Alcoholic liver disease (ALD) develops due to the abuse of ethyl alcohol. The liver, as the main site of ethyl alcohol metabolism, is particularly exposed to the harmful effects of alcohol, and its damage is usually the most visible in the clinical picture. In addition to the damage to the liver itself, changes in many other organs of the body and psychophysical dependence on alcohol are also observed in the patient. The reasons for developing liver damage and its biological mechanisms are not fully understood. It is known that the elimination mechanism of ethyl alcohol from the body consists of its oxidation to aldehyde and acetic acid. This process, in turn, leads to an imbalance of oxidation and reduction processes in the cell and the occurrence of oxidative stress (imbalance between the action of free oxygen radicals and the action of mechanisms that remove them), which is considered the main damaging factor. Depending on the time, amount of alcohol consumed, and genetic conditions, the observed changes in the liver may result in fatty liver, inflammation, or cirrhosis. Individual disease entities do not differ clearly from each other and sometimes occur simultaneously. These are believed to be stage 3 of alcoholic liver damage [120,121,122,123,124]. The contribution of acrolein to hepatocyte cell damage during alcoholic liver disease is shown in Figure 2.

Consuming excessive amounts of alcohol causes the generation of reactive oxygen species, which, among other things, causes lipid peroxidation due to free-radical reactions and the formation of highly reactive acrolein, which plays a significant role in the pathogenesis of ALD [125]. Studies conducted in in vivo animal models and in vitro cell cultures indicate that acrolein is toxic to many cell types [110,126,127,128]. It should be emphasized that most acrolein toxicity mainly affects local tissues and cells directly exposed to it. Therefore, such a biological effect is very important from the point of view of the pathogenesis of many diseases and disorders in which acrolein plays a role [23]. Although the role of acrolein as a pathological mediator in ALD is not fully understood, based on the available literature, it can be concluded that the very high reactivity of the aldehyde and its ability to interact with numerous proteins regulating gene expression, enzymes, and signaling proteins play a key role. The basis of the cytotoxic effect of acrolein in hepatocytes during ALD is undoubtedly the very high ability of this unsaturated aldehyde to disturb the redox balance conducive to the production of ROS. Importantly, as research shows, acrolein-dependent oxidative stress can be realized at various molecular levels and may concern GSH depletion, mitochondrial dysfunction, endoplasmic reticulum stress (ER stress), protein carbonylation, or even disorders of signaling pathways [27,125,129,130,131,132]. One of the primary targets of acrolein is tripeptide glutathione (GHS), which acts as an important free-radical scavenger. Acrolein interacts directly with GSH in hepatocytes, causing its structure modifications and, thus, inactivation [125,129,130,133,134]. Furthermore, as emphasized by the research of Chen et al. [125], acrolein is also responsible for the downregulation of the glutathione-s-transferase-Pi enzyme, which eliminates acrolein through glutathione conjugation under physiological conditions [125]. Interestingly, GSH is not the only molecule involved in maintaining proper redox homeostasis, which is the target of acrolein. As emphasized by Meyers et al. [135,136] and Spiess et al. [137], acrolein is also able to inactivate thioredoxins (Trxs), peroxiredoxins (Prxs), and thioredoxin reductase in respiratory cells.

The consequence of acrolein-dependent inactivation of the GHS, and possibly other systems controlling ROS scavenging, is mitochondrial dysfunction, endoplasmic reticulum stress (ER), and ultimately, hepatocyte damage and apoptosis. Hepatic ER stress and the unfolded protein response (UPR) play a key role in the pathogenesis of ALD [125,129]. Recent studies have indicated that acrolein activates the ER stress pathway in hepatocyte cells by affecting key genes whose protein products involved in cell signaling control the ER stress process. Research by Mohammad et al. [129] demonstrated that acrolein in HepG2 cells upregulated ATF4, ATF3, and Gadd153/CHOP, which are proteins that are critical in cell survival/death in ER stress-associated apoptosis. The consequence of the action of acrolein was also an increase in the expression of caspase-4 and -12, which is directly involved in ER stress. This suggests a strong involvement of ER stress in the hepatotoxic effects of acrolein. It should be emphasized that these data were confirmed by the later work of Chen et al. [125], whose in vitro and in vivo studies also proved a significant contribution of ER stress to acrolein toxicity during ALD. In addition, the consequence of hepatocyte exposure to acrolein manifested by the formation of adducts with proteins (including GSH) and the induction of ER stress is a progressive disruption of mitochondrial integrity/function. In vitro and in vivo studies have shown that exposure to acrolein caused a significant increase in mitochondrial lipid peroxidation and the amount of ROS generated in these organelles. A characteristic feature was a decrease in the activity of enzymes involved in the Krebs cycle and the respiratory chain and, thus, ATP depletion [125,129,130,131,134]. These events ultimately led to acrolein-dependent induction in hepatocytes of activating apoptosis pathways. This was related to the changes in the mitochondrial potential characteristic of this process, the release of cytochrome c from the mitochondria, the increase in the expression of pro-apoptotic proteins of the Bcl-2 family, and the activity of caspase-3 and -9 [125,131].

As emphasized by several groups, acrolein in hepatocytes affects the activation of the pathway of mitogen-activated protein kinases (MAPKs or stress signaling kinases), which include ERK 1/2 (extracellular signal-regulated kinases 1/2), JNK (c-Jun N-terminal kinases) and p38 MAPK [129,131]. The MAPK pathway regulates the activity of many proteins, enzymes, and transcription factors, and thus, has a wide range of biological effects; among others, they affect proliferation and are involved in cell differentiation and apoptosis [138]. The exposure of HepG2 [129] and LO2 [131] liver cells to acrolein led to an increase in the phosphorylation of MAPK kinases and their enhanced activation. The prolonged and persistently active MAPK pathway in cells was stably exposed to acrolein, which, as indicated by Mohammad et al. [129], mediates hepatocyte apoptosis, leading to increased liver damage. Moreover, as suggested by Yin et al. [131], by influencing the MAPK pathway, acrolein modulates the immune response in hepatocytes, leading to an increase in the expression of inflammatory mediator proteins—TNF-α, inducible NO synthase, cyclooxygenase-2 (COX-2), interleukin 1β (IL-1β), and interferon-γ (IFN-γ)—which results in cell damage and eventual death.

Very valuable data on the role of acrolein in the pathogenesis of ALD were provided by the study of Vatsalaya et al. [132]. The study provides a proof of concept for the proposed pathological role of acrolein in ALD. To our knowledge, these are the only studies conducted on patients with ALD. The urinary concentration of the acrolein metabolite (3-hydroxypropyl mercapturic acid) was 4.2 times higher in patients with severe acute alcoholic hepatitis than in patients with mild alcoholic hepatitis. The increase in the acrolein metabolite in patients with ALD corresponded to greater liver damage. In addition, the involvement of acrolein in liver damage was further supported by a strong positive correlation in combination with cytokines involved in the immune response, including Il-1β, IL-8, and the transcription factor TNF-α. The liver is normally resistant to the toxic effects of TNF-α; however, alcohol sensitizes hepatocytes to TNF-mediated apoptosis [132].

## 6. Acrolein as a Carcinogen in Colorectal Cancer

According to IARC, acrolein is classified as a group 2A compound, meaning it is probably carcinogenic to humans. A major limiting factor in studying the direct effects of acrolein on carcinogenesis and mutagenicity is that acrolein levels are difficult to quantify, especially in vivo. Acrolein is rapidly converted to intermediate metabolites or is partially inactivated and detoxified in the human body, mainly due to GSH. However, the participation of acrolein in the process of carcinogenesis is beyond doubt, as recent in vivo and in vitro studies have indicated its high involvement in cancer development [20,139,140]. Squamous cell carcinomas in the nasal cavity were observed in mice and rats that inhaled acrolein for 2 years [141]. Thus far, a significant part of the research devoted to the participation of acrolein in cancer has concerned lung, bladder, and oral cancer. This fact is unsurprising because cigarettes and tobacco products (including e-cigarettes) are one of the main sources of human exposure to this aldehyde. It should be emphasized that acrolein, as a compound with very high reactivity, is extremely toxic in itself, and in addition, it can enhance the toxic effects of other mutagens present in cigarette smoke, e.g., polycyclic aromatic hydrocarbons (PAHs) [142]. In addition, as indicated by recent studies by Hong et al., acrolein, by affecting the HER2/FGFR3 signaling pathways, may contribute to reducing the effectiveness of conventional chemotherapy in the treatment of bladder cancer. The authors emphasized that acrolein in cigarette smoke significantly reduces the sensitivity of cancer cells to cisplatin [143]. Of course, apart from cigarettes, a very important source of exposure to acrolein is our diet, i.e., amino acids, alcohol, carbohydrates, and lipids, including vegetable oils and animal fats, from which aldehyde can be formed during thermal processing. Thus, humans are exposed to acrolein through the consumption of fat-rich foods. Research indicates that a high-fat diet (HFD) may increase the likelihood of developing colorectal cancer (CRC) [9]. A summary of the mutagenic properties of acrolein and the resulting involvement of the aldehyde in colorectal cancer is presented in Figure 3.

Due to its high reactivity, acrolein can interact with a number of proteins in our body, contributing to the inactivation or significant weakening of their functions. The classic example here is the biological inactivation of GHS glutathione, but this is not the only target for acrolein. According to Zarkovic et al. [144], acrolein–protein adducts may be associated with the transition from benign to malignant colorectal tumors. As the authors emphasize, there is a positive correlation between the level of acrolein–protein adducts and the degree of malignancy of the tumor. The presence of acrolein–protein adducts increased based on the degree of colon carcinogenesis. Immunohistochemical studies have indicated moderate levels of acrolein–protein adducts in low-grade adenomas. In comparison, high-grade villous tubular adenomas and Duke’s A carcinomas showed significant amounts of acrolein–protein adducts [144].

Chronic inflammation accompanying the excessive generation of ROS and oxidative stress is one of the important factors in the progression of intestinal cancers. Significant ROS generation is observed in patients with inflammatory bowel disease (IBD), such as Crohn’s disease (CD) and ulcerative colitis (UC). The action of free radicals underlies the pathogenesis of these diseases and the high incidence of colorectal cancer in such patients. One of the important signaling pathways involved in the protection against ROS in the intestines is the nuclear factor erythroid 2-related factor 2 (Nrf2) signaling pathway. Although there are no studies directly indicating the acrolein-dependent weakening of this pathway in the cells of the intestine or the digestive system, several in vitro studies on the cells of the respiratory system indicated that acrolein is able to interact directly with the member of the basic leucine zipper transcription factor family Nrf-2. Thus, the aldehyde can change the upstream and downstream components of the signaling cascade and affect the expression of specific genes [145,146]. Therefore, it cannot be ruled out that acrolein-dependent changes in Nrf-2 activity may also be observed in the intestines and may lead to the progression of intestinal cancers. However, these data should be confirmed in future studies.

Based on the available literature, the carcinogenic molecular activity of acrolein is undoubtedly based on its high reactivity, which allows it to react with, among others, DNA nucleotide bases and acrolein-dependent redox metabolism disorders. The interaction with DNA damages the genetic material and the production of interchain cross-links of double-stranded DNA and DNA-protein cross-links [68] and, consequently, leads to mutations, while redox disturbances lead to oxidative stress and accompanying chronic inflammation. The main target for acrolein is deoxyguanosine (dG), and the resulting acrolein–dG (Acr-dG) adducts are the basis for the G to T transversion and the G to A transition [147]. It is valuable information that recent studies have indicated the diagnostic role of acrolein–DNA adducts in the assessment of cancer risk [139,148,149].

In their in vitro studies, Choudhury et al. [150] analyzed the biological activity of alpha- and beta-unsaturated aldehydes (acrolein) or enals ((E)-4-hydroxy-2-nonenal; HNE) formed as a result of ω-3 and ω-6 polyunsaturated oxidation fatty acids (PUFAs). These compounds play a crucial role in the pathogenesis of colon cancer. Both acrolein and HNE can form cyclic additives of deoxyguanosine in DNA, which have potential mutagenic properties. As emphasized by the authors, Acr-dG adducts were repaired much slower than HNE-dG in HT29 human colon cell extracts. The significantly reduced repair kinetics of Acr-dG are likely due to the NER repair systems’ poor recognition/excision of DNA changes. Moreover, there was a correlation between Acr-dG and HNE-dG adducts—Acr-dG repair was significantly inhibited by HNE-dG. On the other hand, interestingly, no changes in the rate of repair of HNE-dG by Acr-dG were observed [150]. Based on the results of the work of Wang et al. [13] and Lee et al. [151], it can be assumed that acrolein significantly weakens the DNA repair systems associated with base excision repair (BER) and nucleotide excision repair (NER) by inactivating key enzymes involved in these processes.

New light has been shed on the participation of acrolein in colorectal tumorigenesis by the recent, highly valuable research of Tsai et al. [9]. In vitro studies on human normal colon cells, CCD 841 CoN, and the mouse model of fibroblasts NIH/3T3 showed that acrolein-induced oncogenic transformation manifested in an uncontrolled cell cycle, increased profiling, and cell migration, among others. The xenograft tumorigenicity assays indicated that the acrolein-transformed NIH/3T3 clone formed tumors. Moreover, the analysis of samples taken from colorectal cancer patients showed significantly more acrolein-dependent DNA damage (Acr-dG adducts) than in healthy patients. The key to enhanced cell proliferation, colony formation activity, and cell migration was the acrolein-dependent activation of the RAS-MAPK kinase pathway. According to the authors, in cells exposed to acrolein, the upregulation of the Rnd1, Rras2, myc, and PI3Kcb genes was observed [9].

## 7. Concluding Remarks

Human exposure to unsaturated aldehydes remains widespread and results from environmental, internal, personal, and occupational sources. There are several significant sources of unsaturated aldehydes to which we are exposed. They are commonly found in air pollution that is produced significantly during meadow fires, biomass burning, and smoking cigarettes or thermal food processing. Acrolein is an unsaturated aldehyde of greatest concern due to its biological reactivity. What is the impact of exposure to acrolein on human health? The role of acrolein in the pathogenesis and etiology of many diseases is undoubtedly significant. The key to the molecular mechanisms of acrolein in many diseases is the ability to interact with a number of proteins with different functions and their biological impairment, as well as the extremely high activity of generating oxidative stress. As emphasized by numerous studies, the concentration of acrolein or its metabolite is significantly elevated in the urine, plasma, or tissues of patients with many types of diseases, which indicates the participation of acrolein in the initiation and progression of pathological conditions. It is suggested that acrolein affects the pathogenesis of many other diseases, especially those related to the nervous, cardiovascular, respiratory, and digestive systems. The most obvious single source of acrolein is tobacco smoke. The biological activity of acrolein underlies the pathology and development of lung diseases and damage, including COPD and respiratory cancers, and is a serious threat and challenge to modern medicine.

Currently, more attention is being paid to the daily diet and food processing and preparation methods because dietary intake (along with tobacco products) is the main exogenous source of acrolein. Due to its simple yet highly reactive structure, acrolein can quickly interact with micro- and macromolecules in food to form various products, including the very harmful acrylamide. Interference of dietary acrolein in metabolic processes may indirectly modulate the expression of key genes and affect cellular states, which promotes the development of diseases associated with the digestive system. In this review, we focused on discussing the most important molecular mechanisms of acrolein activity known thus far, which are crucial in the pathogenesis of diabetes, alcoholic liver disease, and intestinal cancer. Undoubtedly, several questions regarding the complex mechanisms of acrolein’s molecular activity still need to be answered. Further research is needed to better understand the specific mechanisms of acrolein-induced changes in diabetes or gastrointestinal tumorigenesis. A comprehensive understanding of the mechanisms related to, e.g., acrolein-induced retinal, nephron, or neuronal damage in diabetes and the effects on cell signaling in cancer may help establish acrolein uptake as a novel therapeutic intervention.

## Figures and Tables

**Figure 1 ijms-24-06579-f001:**
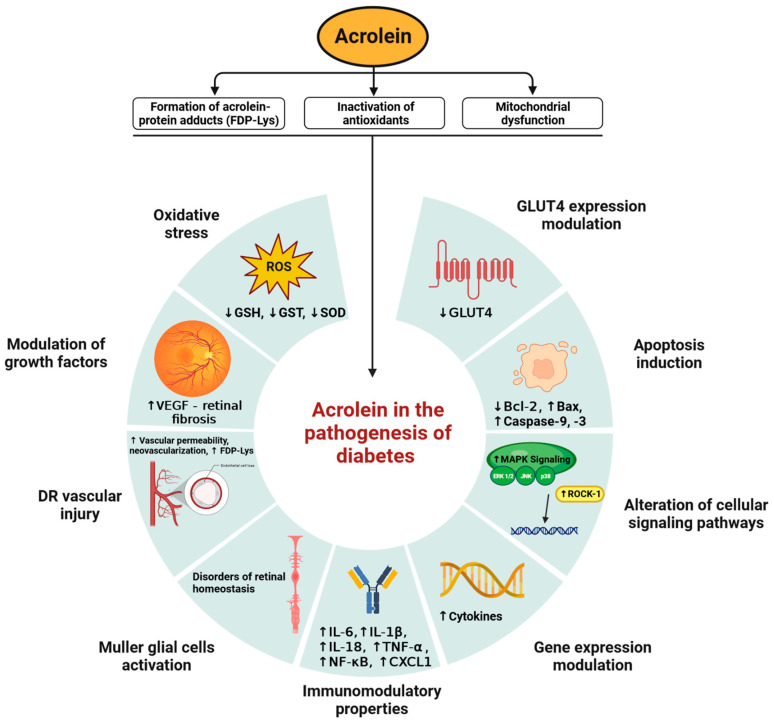
Molecular basis of acrolein participation in the pathogenesis of diabetes. Acrolein, as a highly reactive aldehyde, interacts with several proteins, leading to their inactivation. Antioxidant depletion and damage to mitochondrial function lead to further biological consequences crucial for the development and course of diabetes.

**Figure 2 ijms-24-06579-f002:**
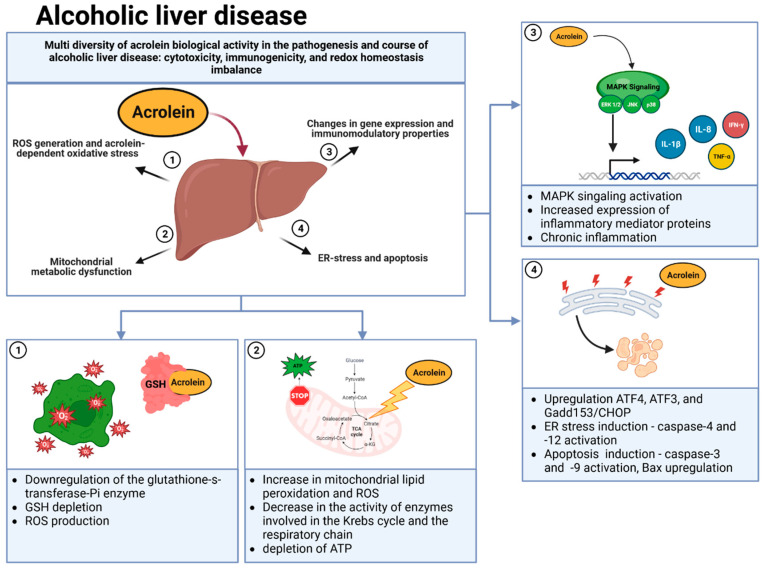
Schematic representation of the potential mechanisms of action of acrolein in the development and progression of alcoholic liver disease: (1) GSH depletion and redox homeostasis disorders, (2) inhibition of mitochondrial biochemical activity, (3) immunogenic and pro-inflammatory properties, (4) induction of ER stress and apoptosis.

**Figure 3 ijms-24-06579-f003:**
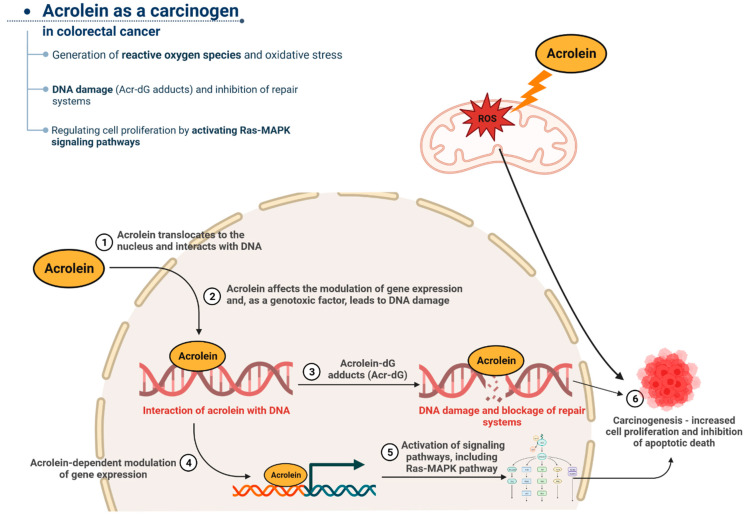
The role of acrolein in the development of colorectal cancer. Acrolein, by interacting with DNA, leads to significant damage and mutation of nitrogenous bases. In addition, aldehyde inhibits repair systems, affecting the fixation of mutations. By activating specific signaling pathways, acrolein induces increased and uncontrolled proliferation of cancer cells. In addition, the consequence of the excessive proliferation of cancer cells is the inhibition of apoptosis. An imbalance between the number of proliferating and dying cells leads to tissue overgrowth characteristic of cancer.

## Data Availability

Data sharing not applicable.

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
