# Peer review of "Diet as a Source of Acrolein: Molecular Basis of Aldehyde Biological Activity in Diabetes and Digestive System Diseases"

_ijms, 2023, doi:10.3390/ijms24076579_

Round 1

Reviewer 1 Report

This review summarized the recent researches on diet as a source of acrolein is interesting. In my opinion, the paper can be published on the IJMS after minor revision. I'd like to ask the author to include line numbers in the MS, as this will be very helpful to the reviewer. 

Author Response

Thank you very much for sending us comments on our paper entitled: “Diet as a source of acrolein: molecular basis of aldehyde biological activity in diabetes and digestive system diseases” (ijms-2308944). We appreciate the time and effort you dedicated to providing feedback on our manuscript. We are grateful for the valuable and inspiring comments, suggestions, and criticisms, which helped improve the paper. 

Reviewer 2 Report

A review manuscript by Hikisz and Jacenik discusses the current knowledge on acrolein and its role in gastrointestinal tract, including both physiological and pathological conditions. The manuscript is well and interestingly written.

Specific comments:

1. The Authors mention that acrolein can be generated from PUFAs, and induces ferroptosis. As ferroptosis is relatively novel pathway of regulated cell death, short notice on its mechanisms/definition should be included along a first mention.

2. Several recent papers in the field should be included, e.g., PMID: 36830667, PMID: 36915964, PMID: 36752871

3. Fig. 2 - panel 3 - (a) it is not necessary to show arrow in circles with IL-8 etc. An arrow showing transcriptional induction is sufficient. In addition, please specify "MAPK signaling" in this figure and show, which MAPK cascade you mean.

4. Fig. 3 - panel 6 - "activation - green light" for cell survival is the same as "stop" for cell death inhibition. Please clarify and show appropriately.

Author Response

(The authors gave the same response as above.)
